# SPECTRALFLOW: GEOMETRY-AWARE MESH ANIMATION VIA SPECTRAL COEFFICIENT DIFFUSION

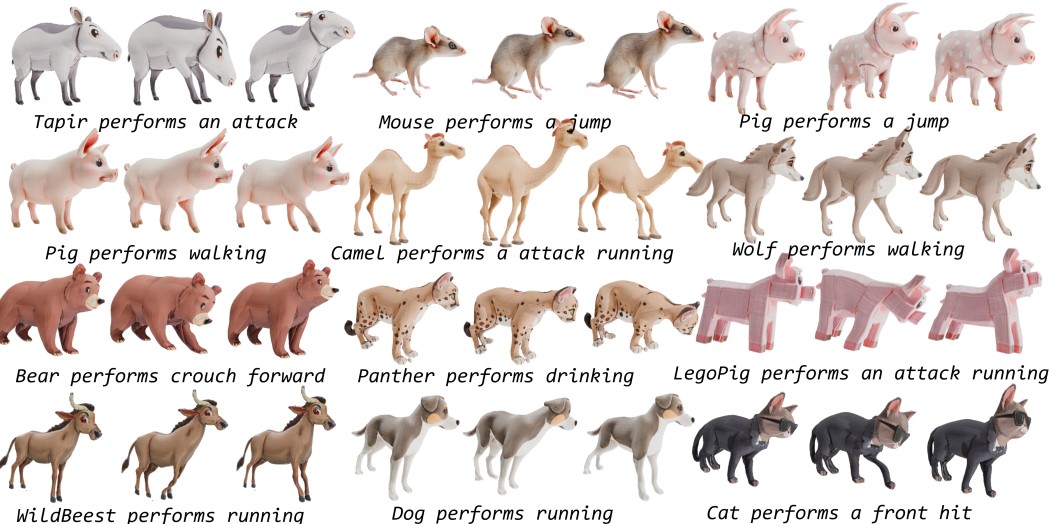

Figure 1: Our **SpectralFlow** generates realistic motion sequences for a variety of 3D shapes, conditioned on action labels (e.g., jump, run, walk, etc.). Animations are generated in **unseen** shapes produced by an image-to-3D model (Li et al., 2024a), demonstrating strong generalization in both categories and actions. The input images are synthesized using Flux (StabilityAI, 2023), a text-to-image diffusion model.

## ABSTRACT

Generating realistic 3D shape sequences (or 4D shapes) conditioned on actions is challenging due to high-dimensional, non-linear, and temporally coherent deformations across diverse shapes. In this work, we introduce SpectralFlow, a diffusion-based framework for action-conditioned 4D shape generation in the Laplacian spectral domain. Instead of modeling raw vertex trajectories or mesh offsets, we represent each shape using a fixed set of Laplacian eigenbases and a sequence of time-varying spectral coefficients, capturing intrinsic geometry and temporal dynamics compactly. By aligning eigenbases across shapes via sign correction and basis transformation, we establish a shared, topology-agnostic spectral space that supports consistent learning across identities and motion types. A conditional diffusion model is trained to generate spectral trajectories based on the input shape and target action, producing smooth, coherent, and semantically aligned mesh sequences. Our method avoids purely implicit modeling, which typically requires large-scale data, by leveraging lightweight geometric representations for controllable 4D shape generation. Extensive experiments show that SpectralFlow outperforms prior methods in reconstruction quality and motion generalization. Our project page is https://specflow3d.github.io.

## 1 INTRODUCTION

Understanding and generating dynamic 3D geometry is a fundamental challenge in computer vision and graphics, with broad applications in animation, robotics, AR/VR, and motion analysis. While static shape modeling has seen great progress, real-world objects are inherently dynamic, requiring

methods that can generate realistic mesh sequences conditioned on motion. Recent advances in 4D shape generation (Zhang et al., 2025; Wu et al., 2025; Jiang et al., 2024) have shown promising results, but progress is limited by the lack of large-scale, high-quality 4D datasets. Unlike the transition from 2D to 3D, modeling temporally evolving 3D meshes demands exponentially more data to capture the diversity of motions per shape, creating a major bottleneck for learning generalizable motion patterns.

Existing methods typically fall into two main categories. Template-based approaches (Loper et al., 2015; Pavlakos et al., 2019; Romero et al., 2017a) assume fixed mesh topologies and are limited in expressiveness for complex or topologically varying motions. On the other hand, implicit neural representations (Park et al., 2019; Zhang et al., 2025; Wu et al., 2025; Jiang et al., 2024) model shape dynamics as continuous fields via coordinate-based MLPs, offering flexibility but often at the cost of generalization or detail preservation. In particular, these approaches tend to suffer from one or more of the following: (**1**) the high dimensionality of geometric data necessitates complex pipelines such as VAE-style encoders or multi-stage training; (**2**) Implicit field-based models, while compact, require substantial data to capture complex 3D shape motions, and often fail to generalize under limited 4D supervision.

To overcome the limitations of purely implicit models, we introduce the explicit geometric structure into 3D dynamic generation. We observe that, for a given shape, the deformations across the motion sequence are nearly isometric, and their Laplacian eigenvectors (Rustamov et al., 2007) remain similar. In our work, we refer to these eigenvectors as a spectral basis, which is shared across the motion sequence to support consistent representation. This allows motion to be represented by varying spectral coefficients over time, providing a structured and data-efficient alternative to fully implicit methods. Modeling motion in the spectral domain offers two key advantages. First, spectral representations exhibit strong consistency across species with similar structures, enabling cross-species motion transfer and generalization. Secondly, alignment in spectral space is significantly easier than in 3D space, as its low-dimensional, compact nature captures deformation effectively while reducing complexity.

In this work, we propose a spectral diffusion framework for action-conditioned shape sequence generation that operates in the spectral domain, rather than modeling raw vertex trajectories or implicit fields. Using Laplacian decomposition, we represent motion as temporal evolution of spectral coefficients. To ensure temporal and cross-species consistency, we align laplacian eigenbasis across shapes via basis transformation and sign correction, enabling learning in a shared, topology-agnostic space. This allows handling arbitrary mesh connectivity without requiring consistent structures. A regularized reconstruction process stabilizes the spectral coefficients, capturing non-rigid deformations faithfully. These aligned coefficients are then fed into a conditional diffusion model, which generates dynamic mesh sequences guided by high-level action semantics. By denoising spectral trajectories conditioned on the input shape and target action, the model produces realistic, coherent, and semantically aligned outputs.

Our lightweight pipeline avoids latent compression and multi-branch designs. Spectral modeling yields smooth motion aligned with text. As shown in Fig 1, it generates realistic, coherent mesh sequences. Experiments on Image-to-3D (Li et al., 2024a) and Animal3D (Xu et al., 2023) confirm strong generalization. In summary, the main contributions of this work are:

- we propose a spectral diffusion framework in a globally aligned spectral coefficient space, enabling generative modeling of 3D mesh sequences with arbitrary and inconsistent topologies; and
- we design a spectral basis alignment and sign correction scheme that establishes a shared spectral space across mesh sequences with varying topology, eliminating the need for topological consistency or training additional encoder-decoder networks; and
- we leverage a diffusion model over the shared spectral space to jointly align action semantics and spectral basis representations, enabling controllable and generalizable sequence generation across diverse identities and species.

## 2 RELATED WORK

**Representation of Deformable Shapes.** Traditional approaches for modeling deformable shapes often rely on parametric or skeleton-based models. For example, SMPL (Loper et al., 2015) and MANO (Romero et al., 2017b) provide compact and robust representations for human bodies and

hands, respectively. Similar modeling techniques have also been extended to animals, such as SMAL (Zuffi et al., 2017), which learns shape and pose variations of quadrupeds from species-specific templates. These methods typically require a species-specific template to model dynamics effectively. Template-free skeleton-based methods (Baran & Popovic, 2007; Xu et al., 2020; Song et al., 2025) model deformations via joint hierarchies and have been widely used for articulated shapes. However, extracting the skeleton and establishing its connectivity is a challenging problem. Moreover, due to the lack of sufficient training data, its generalization ability is limited. Beyond skeleton-based models, recent work (Niemeyer et al., 2019; Palafox et al., 2021; Tang et al., 2021) has explored neural field representations that use MLPs to model 4D deformable shapes in a more flexible and topology-agnostic manner. These methods typically learn continuous shape and motion fields, offering greater expressivity than template-based models. Further, DNF (Zhang et al., 2025) proposes a dictionary-based 4D neural field with compression to model more compact motion representation. However, the scarcity of 4D data constrains their generalization performance.

**4D Generation from 3D Shapes.** With the growing availability of large-scale 3D datasets, such as Objaverse-XL (Deitke et al., 2023), 3D generation techniques have achieved remarkable progress by leveraging scalable implicit models (Li et al., 2024a; Liu et al., 2025; Li et al., 2024b). Building upon these abundant 3D data resources, augmenting static shapes with motion information to produce 4D sequences has become a promising direction. Human-centric 4D generation (Taheri et al., 2020; Tripathi et al., 2023) has benefited greatly from powerful priors and abundant annotations, achieving breakthroughs in human motion generation. However, it has been difficult to extend to other categories. DNF (Zhang et al., 2025) jointly models and generates both motion and shape. Nevertheless, as it relies on shape-specific motion patterns in the training data, its ability to transfer motions to novel shapes remains limited. AnimateAnyMesh (Wu et al., 2025) proposes a novel DyMeshVAE architecture to disentangle spatial and temporal features for high-quality animation generation. Animate3D (Jiang et al., 2024) designs a new spatiotemporal attention module and multi-view video diffusion model for 4D generation. Due to the scarcity of 4D motion data, the generalization performance of these 4D generation methods is greatly weakened. To address this challenge, we fully exploit the information contained in the input 3D geometry and leverage spectral-domain modeling to capture lower-dimensional deformation fields, thereby enabling more efficient and generalizable 4D generation with fewer data requirements.

# 3 PROPOSED METHOD

We propose SpectralFlow, as illustrated in Fig. 2, a diffusion-based framework for generating 3D mesh sequences conditioned on input mesh $V_0$ and action labels $a$, such as text. Distinct from prior approaches that depend solely on data, SpectralFlow incorporates 3D spectral decomposition to model motion in a shape-dependent spectral domain. By leveraging explicit geometric guidance to capture shared motions, it reduces over-reliance on datasets and thereby achieves stronger generalization.

**Problem Definition.** Let $\mathcal{M}_0 = (V_0, F)$ denote the input mesh, where $V_0 \in \mathbb{R}^{n \times 3}$ are vertex positions and $F$ is the face connectivity. The 3D mesh sequence (or 4D mesh) is represented as $\{\mathcal{M}_t\}_{t=1}^{T}$, where each $\mathcal{M}_t = (V_t, F)$ shares topology with $\mathcal{M}_0$ but differs in geometry. To encode mesh geometry in a compact form, we adopt the Laplacian spectral representation. For input mesh $V_0$, we compute the discrete Laplace-Beltrami operator $\mathbf{L} \in \mathbb{R}^{n \times n}$ and the associated mass matrix $\mathbf{M} \in \mathbb{R}^{n \times n}$ (Rustamov et al., 2007). Solving the generalized eigendecomposition problem, i.e., $\mathbf{L}\phi_i = \lambda_i \mathbf{M}\phi_i$, yields a set of orthonormal basis vectors $\phi = [\phi_1, \ldots, \phi_k] \in \mathbb{R}^{n \times k}$, satisfying $\phi^\top \mathbf{M}\phi = \mathbf{I}$, where $\mathbf{I}$ is the identity matrix. These eigenvectors serve as spectral basis functions for the mesh sequences. Each mesh at time $t$ is then projected into the spectral domain using $C_t = \phi^\top \mathbf{M} V_t \in \mathbb{R}^{k \times 3}$, where $C_t$ are the spectral coefficients encoding the shape at time $t$. Our goal is to model the temporal sequence $\{C_t\}_{t=1}^{T}$ conditioned on the input mesh $\mathcal{M}_0$ and an action label $a$. To ensure consistency across subjects and motions, we first align the eigenbases via basis transformation and sign correction, enabling learning in a shared, topology-agnostic spectral space. We then train a conditional diffusion model to generate high-fidelity spectral trajectories $\{C_t\}_{t=1}^{T}$, which are finally projected back to the spatial domain for mesh reconstruction.

## 3.1 SPECTRAL BASIS ALIGNMENT

Due to potential inconsistencies in the spectral basis functions across different meshes, the relationship between the spectral coefficient domain and the motion domain becomes difficult to learn.

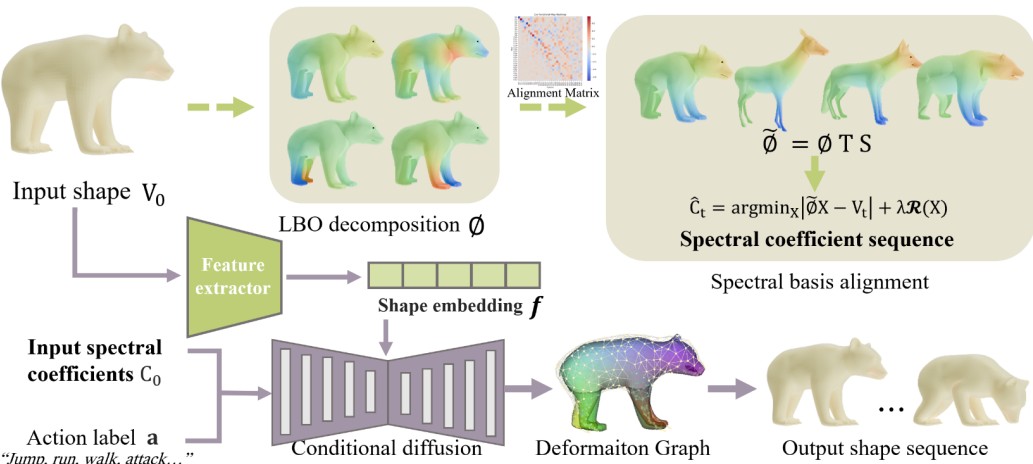

Figure 2: Overview of the proposed SpectralFlow, a novel pipeline for action-conditioned 4D shape generation. Given an input shape $\mathcal{M}_0 = (V_0, F)$ and an action label $\boldsymbol{a}$, we operate in a shared spectral space where spectral basis across shapes are aligned. Motion is modeled as the offset of time-varying spectral coefficients conditioned on the input action. A conditional diffusion model learns to generate coherent spectral trajectories in this aligned space, enabling topology-agnostic shape animation via deformation graph-guided reconstruction from low-dim spectral motion.

Therefore, we solve for a basis transformation that maps the individual basis function into a shared spectral basis domain. Specifically, we construct the spectral basis with the first $k = 32$ low-frequency eigenfunctions, $\phi \in \mathbb{R}^{n \times k}$, which capture the most salient and stable geometric structures of the mesh. The low-frequency modes are more consistent across shapes and primarily capture global geometric and semantic structure, such as pose or coarse deformation. In contrast, high-frequency modes tend to be sensitive to local noise and fine-grained identity details, which are not essential for modeling motion dynamics and may even hinder generalization. To facilitate alignment across sequences, we introduce a reference template with its spectral basis denoted as $\phi_{\text{ref}}$, serving as a canonical coordinate system for all subsequent basis transformations.

**Spectral Basis Transformation.** Given a mesh $\mathcal{M} = (V, F)$ and its truncated eigenbasis $\phi$, we align the Laplacian eigenbases of different shapes into a shared spectral domain, which serves as a canonical space for further processing. This alignment is necessary because the subspace spanned by the first $k$ eigenfunctions may differ across shapes due to variations in orientation, permutation, or other orthogonal transformations within the spectral domain. We transform each shape-specific basis $\phi$ into a shared canonical spectral space defined by the eigenbasis $\phi_{\text{ref}}$ of a selected reference shape. The transformation is modeled by a matrix $\mathbf{T} \in \mathbb{R}^{k \times k}$. We leverage a pre-trained functional map-based feature extractor (Cao et al., 2023) to compute soft pointwise correspondences between each shape and the template shape. From these correspondences, we estimate a functional map $\mathbf{T} \in \mathbb{R}^{k \times k}$ that satisfies $\mathbf{T} \approx \phi_{\text{ref}}^{\dagger} \boldsymbol{\Pi} \phi$, where $\boldsymbol{\Pi} \in [0, 1]^{n_{\text{ref}} \times n}$ is a soft correspondence matrix estimated from a pre-trained descriptor network, with $n_{\text{ref}}$ and $n$ denoting the number of vertices in the reference and input meshes, respectively. Each row sums to 1, representing a probabilistic mapping from source to target vertices. Under the alignment framework with functional map, $\mathbf{T}$ approximates a linear transformation between the eigenspaces of the two shapes. We adopt the estimated transformation matrix $\mathbf{T}$ (as described above) to align the shape-specific basis: $\tilde{\phi} = \phi \mathbf{T}$. This yields an aligned basis $\tilde{\phi}$ expressed in the spectral frame of the reference shape, serving as a canonical domain for consistent comparison and learning.

**Sign Correction for Laplacian Eigenbasis Ambiguity.** To complete the alignment, we further address the sign ambiguity inherent in spectral representations. Spectral representations suffer from inherent ambiguities due to the nature of eigendecomposition, especially when applied across multiple shapes. While this ambiguity has little impact on correspondence tasks, it becomes critical when using eigenfunctions as input features for learning-based models. In such settings, inconsistent signs can confuse the network and hinder convergence, as the same geometric structure may be represented with opposite signs across different shapes (Cao et al., 2023; Zhuravlev et al., 2025).

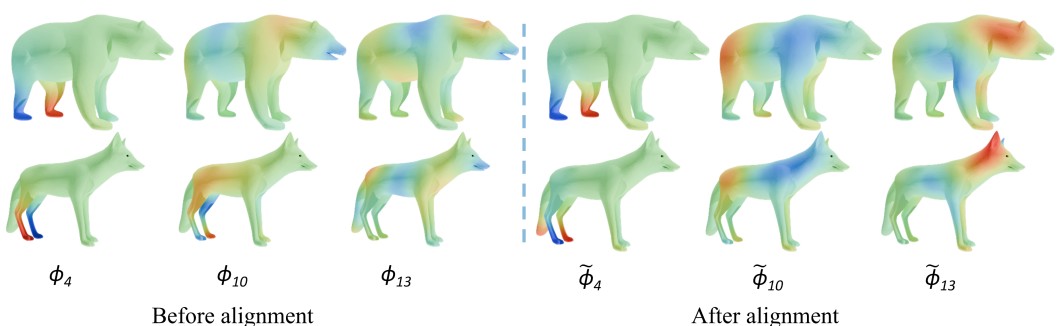

$$\phi_4 \qquad \phi_{10} \qquad \phi_{13} \qquad \widetilde{\phi}_4 \qquad \widetilde{\phi}_{10} \qquad \widetilde{\phi}_{13}$$

Before alignment                    After alignment

Figure 3: Visualization of the aligned basis function $\widetilde{\phi}_i$ on shapes across different species. The corresponding spectral basis functions exhibit strong consistency across shapes, ensuring the comparability of spectral coefficients.

To resolve the sign ambiguity inherent in eigenfunction computation, we construct a diagonal sign correction matrix $\mathbf{S} \in \{\pm 1\}^{k \times k}$ to enforce consistent eigenfunction orientations across shapes. We estimate a set of matched vertex pairs $\mathcal{P} \subset \mathcal{V} \times \mathcal{V}_{\text{ref}}$ between a shape and the reference template from a point-to-point map computed via functional map inversion (Cao et al., 2023). Each pair $(p, q)$ consists of a vertex $p \in \mathcal{V}$ on the current shape and its corresponding vertex $q \in \mathcal{V}_{\text{ref}}$ on the template. We determine the sign of each eigenfunction by comparing its values at these corresponding vertices. Specifically, for each eigenfunction index $i \in \{1, \ldots, k\}$, we compute a sign agreement score:

$$s_i = \text{sign}\left( \sum_{(p,q) \in \mathcal{P}} \phi_i(p) \cdot \phi_i^{\text{ref}}(q) \right), \tag{1}$$

where $\phi_i$ and $\phi_i^{\text{ref}}$ denote the $i$-th eigenfunction of the current shape and the reference shape, respectively. The intuition is that if the aggregated inner product between corresponding eigenfunction values is negative, one of them should be flipped to ensure a consistent orientation.

Denote by $\mathbf{S} = \text{diag}(s_1, \ldots, s_k)$ the sign correction matrix. The aligned basis is further corrected as $\check{\phi} = \widetilde{\phi}\,\mathbf{S}$. This step ensures that the eigenfunctions are consistently oriented across all shapes, providing a stable basis for learning-based models, where sign inconsistency can otherwise confuse the network and degrade performance. After applying both spectral alignment and sign correction, each shape is represented in an orientation-consistent and shared spectral space. As illustrated in Fig. 3, spectral basis alignment and sign correction leads to visibly aligned eigenfunctions across shapes. A detailed cross-species analysis of semantic motion alignment is provided in Appendix A.2, where spectral alignment is shown to significantly enhance the consistency of action trajectories across diverse species.

## 3.2 REGULARIZED SPECTRAL RECONSTRUCTION

After aligning the spectral basis across shapes, our goal is to model 4D motion in the spectral coefficient space. We represent the spectral coefficients of a shape at time $t$ as an offset from an identity-specific anchor $C^*$, which corresponds to the same shape in a canonical (rest) pose and provides a stable reference point for temporal modeling:

$$\Delta C_t = C_t - C^*, \tag{2}$$

where $C_t \in \mathbb{R}^{k \times 3}$ denotes the spectral coefficients at time $t$, and $C^*$ represents the spectral coefficients of the same identity in a rest (i.e., undeformed) pose. This formulation allows the model to learn identity-specific deformations relative to a canonical pose, thereby improving generalization across motion sequences and categories.

However, aligning shapes across species introduces additional challenges. Due to anatomical and topological differences, the alignment matrix $\mathbf{T}$ can become ill-conditioned or nearly singular. This instability amplifies small variations in geometry into large fluctuations in the resulting spectral coefficients, degrading robustness and interpretability.

To address this, we propose to recover the spectral coefficients not by direct projection, but by solving a regularized optimization problem. Given observed vertex positions $V_{\text{target}} \in \mathbb{R}^{n \times 3}$ at time $t$, we solve for spectral coefficients $\hat{C} \in \mathbb{R}^{k \times 3}$ that minimize the following energy:

$$\mathcal{L}_{\text{recon}}(\hat{C}) = \underbrace{\|\mathbf{M}^{1/2}(\tilde{\phi}\hat{C} - V_{\text{target}})\|_F^2}_{\text{Data fidelity}} + \underbrace{\lambda_{\text{Lap}}\|\mathbf{L}(\tilde{\phi}\hat{C})\|_F^2}_{\text{Laplacian smoothness}} + \underbrace{\lambda_{\ell_2}\|\hat{C}\|_F^2}_{\text{L2 regularization}}, \tag{3}$$

where $\mathbf{M}$ is the mass matrix, and $\mathbf{L}$ is the discrete Laplacian operator. The first term ensures reconstruction accuracy under a mass-weighted metric. The second term encourages the reconstructed geometry to be spatially smooth, while the third controls the magnitude of the coefficients and prevents overfitting.

**Efficient and Robust Closed-Form Spectral Reconstruction.** Minimizing the quadratic objective in Eq. (3) leads to a closed-form solution via the normal equations:

$$\mathbf{A}_{\text{reg}}\,\hat{C} = \mathbf{b}, \quad \text{where} \quad \begin{cases} \mathbf{A}_{\text{reg}} = \tilde{\phi}^\top \mathbf{M}\tilde{\phi} + \lambda_{\text{Lap}}\tilde{\phi}^\top \mathbf{L}^\top \mathbf{L}\tilde{\phi} + \lambda_{\ell_2}\mathbf{I} \\ \mathbf{b} = \tilde{\phi}^\top \mathbf{M}V_{\text{target}}. \end{cases} \tag{4}$$

The solution $\hat{C}$ corresponds to the estimated spectral coefficients that best reconstruct the target shape under both data fidelity and regularization constraints. This formulation ensures both efficiency and robustness. Since the system matrix $\mathbf{A}_{\text{reg}}$ is constant across time steps and independent of the target shape, it can be precomputed and factorized (e.g., via Cholesky decomposition) once per sequence. The inclusion of the Laplacian and $\ell_2$ terms stabilizes the solution, making it resilient to noise, missing vertices, or ill-conditioned basis transformations—particularly important in cross-species scenarios.

Once recovered, the shape at time $t$ is reconstructed as:

$$\widehat{V}_t = \tilde{\phi}(\hat{C}^* + \Delta\hat{C}_t), \tag{5}$$

where $\tilde{\phi}$ denotes the aligned spectral basis shared across all frames of the same shape, $\Delta\hat{C}_t$ is the estimated spectral offset, and $\hat{C}^*$ denotes the estimated spectral coefficients of the shape in a canonical pose, serving as an anchor for temporal modeling. This enables the model to decode temporally evolving shapes from a stable and semantically aligned spectral space.

### 3.3 Diffusion-based Spectral Coefficient Modeling

To model the temporal evolution of 3D shape deformation in the spectral domain, we employ a conditional denoising diffusion probabilistic model (DDPM) (Ho et al., 2020), which generates a sequence of time-dependent spectral coefficient offsets $\{\Delta\hat{C}_1, \ldots, \Delta\hat{C}_T\}$. Each offset $\Delta\hat{C}_t \in \mathbb{R}^{k \times 3}$ is added to $\hat{C}^*$, the spectral coefficient vector in a canonical pose, to reconstruct the corresponding frame, as defined in Eq. (5). The base coefficient $\hat{C}^*$ encodes the static shape identity (e.g., the neutral pose), while $\hat{C}_t$ captures temporally-varying deformations (e.g., body movement or expression changes). Modeling in the spectral domain provides a compact and semantically aligned representation that facilitates learning smooth and consistent temporal dynamics.

**Modeling Temporal Spectral Offsets via Diffusion.** Given a sequence of spectral coefficient offsets $\{\Delta\hat{C}_t\}_{t=1}^T \in \mathbb{R}^{T \times k \times 3}$, we first map the 3D offsets into a $d$-dimensional feature space via a linear projection, resulting in features $\mathbf{x} \in \mathbb{R}^{T \times k \times d}$. We then flatten the spectral dimension to form per-frame tokens $\mathbf{x} \in \mathbb{R}^{T \times (k \cdot d)}$, and add learned positional embeddings to encode temporal order. This token sequence serves as the input to a Transformer, which models long-range temporal dependencies across frames. To learn a generative model over these temporal spectral deformations, we adopt a diffusion probabilistic model (DDPM) to model the distribution of offsets $\{\Delta\hat{C}_t\}_{t=1}^T$, defined with respect to the input coefficients $\hat{C}_0$. At each denoising timestep $t$, the ground-truth offset $\Delta\hat{C}_t$ is perturbed by Gaussian noise:

$$\Delta\widetilde{C}_t = \sqrt{\bar{\alpha}_t}\,\Delta\hat{C}_t + \sqrt{1 - \bar{\alpha}_t}\,\epsilon, \quad \epsilon \sim \mathcal{N}(0, I), \tag{6}$$

and train a denoising network $\epsilon_\theta$ to recover the noise. The network is conditioned on the clean first-frame coefficient $\hat{C}_0$, the current timestep $t$, and additional semantic $\boldsymbol{a}$ and geometric features $\boldsymbol{f}$. The training objective is

Figure 4: Animation examples of SpectralFlow. Our method generates realistic and coherent motion generation for meshes obtained from image-to-3D (Li et al., 2024a).

$$\mathcal{L}_{\text{diff}} = \mathbb{E}_{t, \Delta C, \epsilon} \left[ \left\| \epsilon_\theta([\Delta \widetilde{C}_t, \hat{C}_0], t, \boldsymbol{a}, \boldsymbol{f}) - \epsilon \right\|_2^2 \right], \tag{7}$$

where $\epsilon_\theta$ denotes the denoising network. This formulation enables the model to capture temporally coherent deformation trajectories while maintaining a consistent reference anchored at the first frame.

The textual condition is provided as a one-hot vector $\boldsymbol{a} \in \{0, 1\}^{N_a}$ representing the action label, where $N_a$ denotes the number of motion categories. This encoding is projected to a latent space via an embedding layer and used to guide the generation process. The shape condition $\boldsymbol{f}$ is computed from a canonical shape using DiffusionNet (Sharp et al., 2022). Specifically, given a shape $\mathcal{M} = (V, F)$, we extract per-vertex features $F_x \in \mathbb{R}^{n \times C}$ and project them into the spectral domain using the aligned spectral basis $\tilde{\phi} \in \mathbb{R}^{n \times k}$:

$$\mathbf{Z} = \phi^\top F_x \in \mathbb{R}^{k \times c}, \tag{8}$$

where $k$ is the number of spectral modes and $c$ is the feature dimension, set to 128 in our implementation following the output dimensionality of DiffusionNet. To align the features with the canonical basis of a template shape $\phi_{\text{ref}}$, we apply a functional map $T \in \mathbb{R}^{k \times k}$ and a diagonal sign correction matrix $\mathbf{S} \in \{\pm 1\}^{k \times k}$ obtained by minimizing the sign disparity between $\phi$ and the mapped basis $\phi \mathbf{T}$. The final shape feature is computed as

$$\boldsymbol{f} = \mathbf{S}^\top \mathbf{T}^\top \mathbf{Z} \in \mathbb{R}^{k \times c}. \tag{9}$$

This shape descriptor captures intrinsic geometric structure in a basis-invariant space and is used to condition the diffusion model via cross attention.

The overall training loss combines the standard diffusion objective with a temporal smoothness regularization that encourages coherent motion across time:

$$\mathcal{L} = \mathcal{L}_{\text{diff}} + \lambda_{\text{smooth}} \mathcal{L}_{\text{smooth}}, \tag{10}$$

where $\mathcal{L}_{\text{diff}}$ is the diffusion loss defined in Eq. (7), and $\mathcal{L}_{\text{smooth}}$ is a temporal regularization term applied to the predicted spectral offset sequence $\{\Delta \hat{C}_t\}_{t=1}^T$. Specifically, we define:

$$\mathcal{L}_{\text{smooth}} = \alpha \cdot \mathbb{E} \left[ \left\| \Delta \hat{C}_{t+1} - \Delta \hat{C}_t \right\|_2^2 \right] + \beta \cdot \mathbb{E} \left[ \left\| \Delta \hat{C}_{t+1} - 2\Delta \hat{C}_t + \Delta \hat{C}_{t-1} \right\|_2^2 \right], \tag{11}$$

where the first term penalizes high temporal velocity (encouraging smooth transitions), and the second term penalizes high temporal acceleration (ensuring motion consistency and avoiding sudden jerks). Hyperparameters $\alpha$ and $\beta$ control the relative strength of each term, and $\lambda_{\text{smooth}}$ balances the overall regularization.

**Deformation Graph-Guided High-Resolution Reconstruction.** At inference time, a sequence of spectral offsets $\{\Delta \hat{C}_t\}_{t=1}^T$ is progressively sampled from Gaussian noise via the reverse diffusion process. These offsets are conditioned on the input spectral coefficients $\hat{C}_0$, which are computed from the first frame using the aligned spectral basis. The full spectral sequence is then formed by

Table 1: Quantitative comparison on video generation metrics. "↑" means the larger, the better.

| Method | Image-to-3D | | | Animal3D | | |
|---|---|---|---|---|---|---|
| | I2V ↑ | Mo. Sm. ↑ | Aest. Q. ↑ | I2V ↑ | Mo. Sm. ↑ | Aest. Q. ↑ |
| Animate3d (Jiang et al., 2024) | 0.9295 | **0.9933** | 0.5418 | 0.9603 | 0.9909 | 0.4736 |
| AnimateAnyMesh (Wu et al., 2025) | 0.9493 | 0.9786 | 0.5675 | 0.9178 | 0.9878 | 0.4850 |
| SpectralFlow (**Ours**) | **0.9638** | 0.9922 | **0.5921** | **0.9745** | **0.9956** | **0.5212** |

combining $\hat{C}_0$ with the predicted offsets, and subsequently decoded through the shared basis to reconstruct the 3D mesh sequence $\{\widehat{V}_t\}_{t=1}^T$. Despite reconstructing only low-frequency components, the resulting sequence effectively captures the global motion and semantic structure of the target action. To further enhance the spatial fidelity of these coarse reconstructions, especially under challenging conditions such as noise, occlusion, or partial observations, we introduce a structure-aware motion propagation step based on an embedded deformation graph (Sumner et al., 2007). This graph is constructed on the high-quality mesh of the first frame and consists of a sparse set of anchor nodes, each associated with a local transformation and per-vertex influence weights. We propagate the low-frequency motion sequence $\{\widehat{V}_t\}_{t=1}^T$ by updating the node transformations over time. This yields a refined sequence of deformed meshes $\{V_t^{\text{deformed}}\}_{t=1}^T$ that are both spatially coherent and temporally smooth. See Appendix A.1 for details of deformation graph.

## 4 EXPERIMENTS

**Datasets.** We trained our model on the DeformingThings4D (DT4D) (Li et al., 2023) dataset, which comprised approximately 1,700 animated animal sequences across 50 categories. A notable characteristic of this dataset is that the same type of motion (e.g., walking, jumping) is performed by multiple animal species. This setup enabled the model to learn motion patterns that were decoupled from specific object geometries. For evaluation, we constructed a comprehensive benchmark using two additional datasets. First, we constructed a synthetic evaluation set consisting of 100 unique objects, derived from the image-to-3D generation pipeline (Li et al., 2024a), which produced static 3D meshes from single-view images. These single-view images were synthesized from text prompts using the Flux text-to-image diffusion model (StabilityAI, 2023). Second, we employed the test subset of the Animal3D dataset (Xu et al., 2023), which consisted of 120 real-world animal meshes reconstructed from natural images in ImageNet (Deng et al., 2009). These meshes were generated using a parameterized mesh extraction method that leveraged learned 3D priors and species-specific templates to recover high-fidelity geometry from in-the-wild imagery. Both evaluation datasets contained entirely **unseen shapes**, offering a rigorous testbed for evaluating the shape generalization capabilities of our model.

**Implementation Details.** We trained the model for 150 epochs using the AdamW optimizer with a learning rate of $2 \times 10^{-4}$ and weight decay values of 0.01. Training was conducted on 5 NVIDIA RTX A6000 GPUs and typically converges within 18 hours. During training, we uniformly sampled 25 frames per sequence at a temporal sampling rate of 1. The weights $\lambda_{\text{smooth}}$ in Eq. 7, $\lambda_{\text{lap}}$, and $\lambda_{\ell_2}$ in Eq. 3 were set to $10^{-1}$, $10^{-3}$, and $10^{-3}$, respectively. All meshes were normalized to fit inside a unit sphere centered at the origin.

**Evaluation Metrics.** We adopted the evaluation protocol proposed in VBench (Huang et al., 2024), a widely used benchmark for video generation, following the evaluation setting used in Animate-A-Story (Jiang et al., 2024). Specifically, we used three image-to-video (I2V) evaluation metrics: I2V Subject, Motion Smoothness, and Aesthetic Quality. These metrics respectively assessed the consistency between the generated video and the input image, the temporal smoothness of the motion, and the perceptual quality of the appearance. In our experiment section, we abbreviated them as "I2V", "M. Sm.", and "Aest. Q.". All metrics follow the convention that higher values indicate better performance.

**Quantitative Comparison.** We conducted quantitative evaluations on a curated test set of animal meshes, using the same set of motion prompts for all baseline methods to ensure fair comparison. Following the VBench protocol (Huang et al., 2024), we adopted three evaluation metrics: I2V Subject, Motion Smoothness, and Aesthetic Quality, which respectively assess semantic fidelity to the input mesh, temporal consistency, and visual quality. As shown in Tab. 1, our SpectralFlow outperforms existing approaches, including Animate3D (Jiang et al., 2024) and AnimAnyMesh (Wu

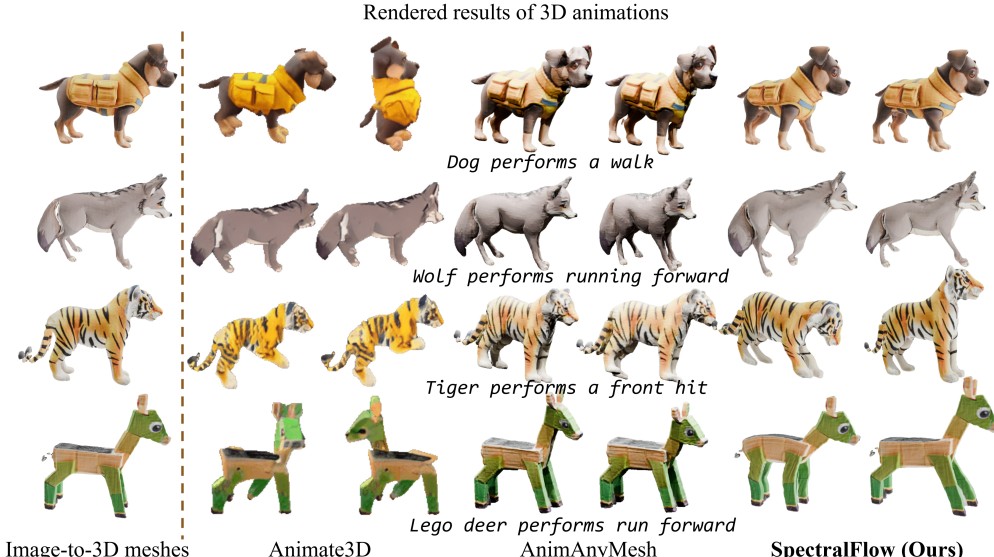

Rendered results of 3D animations

*Dog performs a walk*

*Wolf performs running forward*

*Tiger performs a front hit*

*Lego deer performs run forward*

Image-to-3D meshes     Animate3D     AnimAnyMesh     **SpectralFlow (Ours)**

Figure 5: Qualitative comparison with state-of-the-arts. Our method preserves motion semantics across diverse shapes by learning shared motion representations that generalize across species.

et al., 2025), in most of the three metrics. In particular, we observed significant improvements in I2V and Aesthetic Quality, indicating better preservation of shape details and more appealing appearance. Although Animate3d achieves slightly higher Motion Smoothness on Animal3D evaluation dataset, it exhibits noticeably lower text-motion alignment, as confirmed by our user study (see Tab. 3). Overall, our method achieves a favorable balance between motion dynamics and visual quality, enabling high-quality and semantically meaningful animations for animals. The quantitative results are in strong agreement with our qualitative observations and user study outcomes.

**Qualitative Comparison.** As illustrated in Fig. 5, we presented visual comparisons of animation results generated by different methods on a variety of animal meshes. Existing approaches exhibit noticeable limitations in either appearance fidelity, motion dynamics, or multi-view consistency. As shown in the first and last rows of Fig. 5, Animate3D suffers from severe multi-view motion inconsistency. Its multi-view training strategy often causes temporal and spatial motion mismatches, resulting in shape distortions and reduced geometric plausibility. AnimAnyMesh generates sequences with very limited motion, typically restricted to global body rotation or subtle floating, which results in low alignment between the generated motion and the input text prompt. This issue is also evident in the user study results (see Tab. 3). The underlying cause lies in its motion generation strategy, which directly controls vertex trajectories. The method balances motion expressiveness and geometric stability,

but struggles with complex deformations and shape consistency. In contrast, our SpectralFlow achieves prompt-aligned, temporally coherent animations with detailed and consistent geometry (Fig. 4), outperforming all baselines (Fig. 5). This is attributed to our spectral representation based on explicit geometry. Additional visualizations of our method and more comparisons with other approaches are provided in Appendix A.5 and Appendix A.4, respectively.

**Ablation Study.** We conducted ablation experiments to evaluate the effect of key components and settings in our framework. First, we analyzed the effect of the number of spectral basis functions $k$ used to represent mesh geometry. As shown in

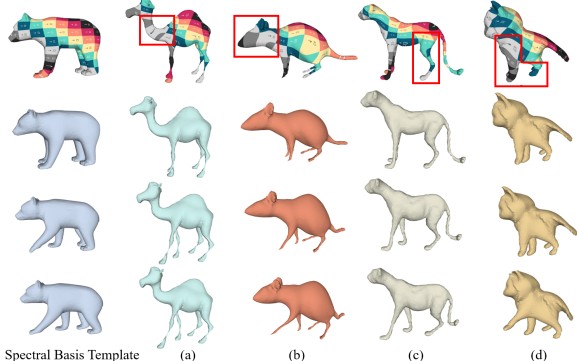

Spectral Basis Template    (a)    (b)    (c)    (d)

Figure 6: Examples of weak and partial correspondences relative to the spectral basis template (red box). Motion transfer remains feasible due to the robustness of low-dimensional spectral matching.

Table 2: Ablation study. Lower values indicate better performance across all metrics.

| Metrics | k=16 | k=64 | Full Vetex | 25%Data | 50%Data | Ours |
|---|---|---|---|---|---|---|
| Diffusion Loss↓ | 0.0085 | 0.0092 | 0.7860 | 0.0263 | 0.0150 | **0.0062** |
| Vetex Error↓ | 0.1466 | 0.1850 | 0.3474 | 0.1262 | 0.1049 | **0.0583** |
| Smoothness Loss↓ | 0.0178 | 0.0169 | 28.560 | 0.0221 | 0.0180 | **0.0101** |

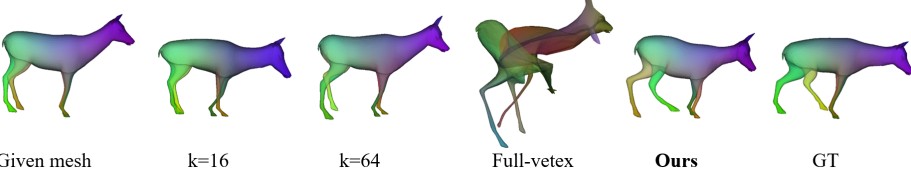

| Given mesh | k=16 | k=64 | Full-vetex | **Ours** | GT |

Figure 7: Visualizations of ablation study. Motion corresponds to the most deformed frame from a new motion sequence on a seen shape. The generated new motion is not identical to the ground truth but conduct the same semantic action.

Tab. 2, employing too few spectral bases (e.g., $k = 16$) results in noticeable degradation in shape reconstruction, as the low-dimensional representation lacks sufficient geometric detail. Conversely, increasing $k$ to 64 negatively impacts motion learning, as the model tends to overfit to high-frequency shape variations and loses focus on the underlying motion dynamics. Conversely, increasing $k$ to 64 negatively impacts motion learning, as the model tends to overfit to high-frequency shape variations, resulting in less clear and coherent motion as shown in Fig. 7. Increasing $k$ introduces greater difficulty in spectral basis matching, especially across diverse topologies. Our default setting of $k = 32$ provides a favorable balance, capturing essential motion while preserving geometric fidelity. We further evaluated a baseline that removes the spectral representation and directly learns from vertex positions. This baseline fails to model coherent motion and produces unstable sequences with poor shape recovery. The lack of structural guidance from spectral representation makes it difficult for the network to learn meaningful temporal patterns or maintain consistency across frames. In addition, we evaluated the effect of training data size by reducing the dataset to 50% and 25% of its original volume. As expected, performance drops slightly with reduced data, particularly in terms of motion smoothness and fine-grained geometry. However, even with only 25% of the training data, the model maintains a level of performance that remains comparable to the full-data setting, indicating that our spectral representation is relatively data-efficient. This is attributed to our spectral representation, which captures shared semantic motion structures across shapes and categories, effectively reducing the learning burden and enabling the model to generalize motion patterns from limited data.

**Robustness to Correspondences.** Instead of requiring accurate point-wise correspondences, our method leverages low-frequency spectral alignment, offering greater robustness and efficiency. As illustrated in Fig. 6, we evaluate our method on challenging cases involving weak and partial correspondences (e.g., (a) the neck of a camel, (b) the head of a mouse, (c) missing limbs and (d) the chest and forelegs of a cloaked cat). These include non-isometric shapes, local mismatches, and meshes with incomplete geometry. Our method is robust to correspondence errors, as successful motion generation only requires approximate alignment in the low-frequency spectral space.

## 5 CONCLUSION AND DISCUSSION

We propose SpectralFlow, a topology-agnostic framework for the generation of 3D mesh sequences. By aligning spectral bases across arbitrary shapes through sign correction and spectral transformation, SpectralFlow constructs a globally consistent coefficient space. Built on this space, a diffusion-based model captures shared action semantics and enables controllable, structure-aware generation conditioned on input shape and action labels. SpectralFlow unifies spectral geometry and generative modeling for dynamic animal motion, enabling editing, cross-species animation, and simulation.

The 4D generation task remains highly challenging due to the current severe scarcity of 4D data. While SpectralFlow has achieved consistent and structure-aware motion generation across various species and significantly outperforms state-of-the-art approaches, it still cannot generalize to arbitrary species or broader motion types. Moreover, we only modeled low-dimensional motion representations to better capture common motion patterns from relatively small datasets. As dataset scales increase, modeling personalized high-frequency details and generating longer, more complex motion sequences would represent a practical and interesting direction for future work.

## STATEMENT

### ETHICS STATEMENT

We adhere to the ICLR Code of Ethics in the conduct of this research. Our work focuses on topology-agnostic 3D mesh animation through spectral basis alignment and generative modeling. The datasets used in this study are either publicly available or synthetically generated using publicly released models, without involving any private, sensitive, or proprietary information. This research is purely technical in nature, and we do not anticipate any ethical concerns or potential for harm.

### REPRODUCIBILITY STATEMENT

We have described the methodological steps in detail in Sec. 3, and elaborated on the implementation, evaluation details and data processing in Sec. 4 and Appendix A.7. The datasets used in our experiments include both publicly available data and synthetic samples generated using publicly released models, and all sources are properly cited. All experiments were conducted on a workstation equipped with an Intel Xeon 4309Y CPU and an NVIDIA RTX A6000 GPU, using PyTorch 2.2 and CUDA 12. Our code, along with the testing data and instructions for data preprocessing, will be open-sourced in the future, thus ensuring strong reproducibility.

### STATEMENT OF AI USE

We used large language models (LLMs), specifically ChatGPT, solely for grammar polishing of the manuscript. All LLM outputs were manually reviewed and verified for accuracy; no content was directly adopted without validation. The authors bear full responsibility for all content presented in the paper.

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

# A APPENDIX

## A.1 DEFORMATION GRAPH-GUIDED RECONSTRUCTION

To propagate the coarse motion sequence to the target mesh with higher spatial fidelity, we adopt a node-graph based deformation model inspired by embedded deformation graph (Sumner et al., 2007). A sparse set of $m$ graph nodes $\{\mathbf{g}_j\}_{j=1}^m$ is uniformly sampled from the input mesh $V_0$ and each node is associated with a local rigid transformation $(\mathbf{R}_j, \mathbf{t}_j)$.

Given a vertex $\mathbf{x}_i$ on the shape surface, we identify its $k$ nearest graph nodes in terms of geodesic distance over the mesh surface, denoted by the index set $\mathcal{N}(i)$. The deformed position $\mathbf{y}_i$ is computed as a weighted sum of locally transformed positions:

$$\mathbf{y}_i = \sum_{j \in \mathcal{N}(i)} w_{ij} \left[ \mathbf{R}_j(\mathbf{x}_i - \mathbf{g}_j) + \mathbf{g}_j + \mathbf{t}_j \right], \tag{12}$$

where $w_{ij}$ is the influence weight of control node $j$ on vertex $i$, satisfying $\sum_{j \in \mathcal{N}(i)} w_{ij} = 1$.

**Gaussian Weighting.** In the default setting, weights are computed using a Gaussian kernel on geodesic distances, followed by softmax normalization:

$$w_{ij} = \frac{\exp\left(-\frac{d_{\text{geo}}(\mathbf{x}_i, \mathbf{g}_j)^2}{2\sigma^2}\right)}{\sum_{j' \in \mathcal{N}(i)} \exp\left(-\frac{d_{\text{geo}}(\mathbf{x}_i, \mathbf{g}_{j'})^2}{2\sigma^2}\right)}, \tag{13}$$

where $d_{\text{geo}}(\cdot, \cdot)$ denotes the geodesic distance computed along the mesh surface.

**Rigid Transform Estimation.** We estimate the rigid transformation of each graph node by the low-frequency reconstruction $\{\widehat{V}_t\}_{t=1}^T$ of the mesh sequence. Since the generated low-frequency meshes in a sequence are topologically consistent, we can estimate the rigid transformation $(\mathbf{R}_j, \mathbf{t}_j)$ of each graph node via SVD-based alignment between its initial position $\hat{\mathbf{g}}_j^{(0)}$ and its position in the current frame $\hat{\mathbf{g}}_j^{(t)}$. Here $\hat{\mathbf{g}}_j \in \widehat{V}$ is the corresponding point of $\mathbf{g}_j$. This strategy enables stable and semantically meaningful alignment even under non-rigid deformations or partial correspondence.

## A.2 ANALYSIS OF SHAPE FEATURES AND MOTION COMMONALITIES

In this section, we present a visualization of the spectral coefficient trajectories corresponding to the semantic action run across multiple species, both before and after spectral basis alignment (Fig. 8). Specifically, the unaligned trajectories are shown in Fig. 8a, while the aligned trajectories are presented in Fig. 8b. To facilitate comparison, the high-dimensional trajectories are projected into two dimensions using Principal Component Analysis (PCA), enabling an intuitive examination of the effect of alignment on cross-species spectral consistency.

The results reveal that, prior to alignment, the trajectories are highly scattered and exhibit no coherent structure, indicating limited similarity in the latent spectral space across species. In contrast, after alignment, the trajectories demonstrate clear structural regularities and become more congruent across different species, suggesting that the alignment procedure effectively captures shared dynamic patterns associated with the run action.

Furthermore, instances belonging to the same species (e.g., Bear3EP, Bear84Q, Bear9AK) form more compact clusters in the aligned space, highlighting the method's ability to enhance intra-species consistency. Importantly, even between distinct species such as moose and deer, the aligned trajectories follow similar paths, providing additional evidence of cross-species spectral commonality resulting from the alignment process.

## A.3 USER STUDY

In addition to the quantitative evaluation, we conducted a user study to assess the perceptual quality and semantic alignment of the generated motions. Participants consistently favored our results,

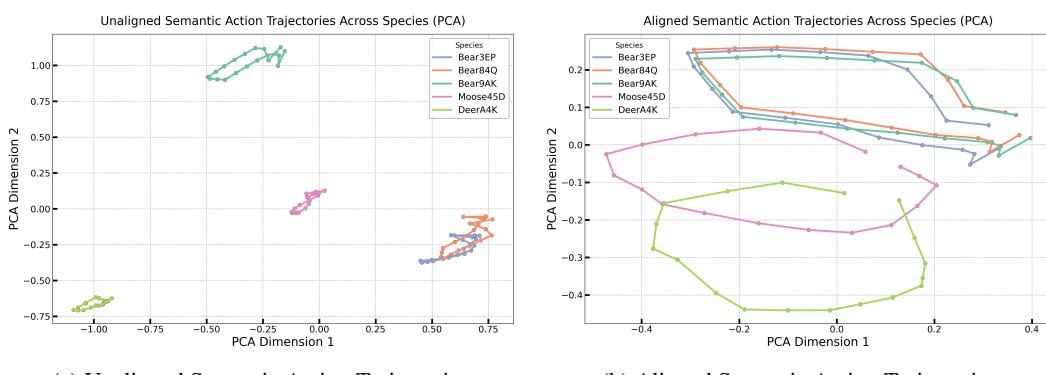

(a) Unaligned Semantic Action Trajectories      (b) Aligned Semantic Action Trajectories

Figure 8: Comparison of spectral coefficient trajectories (across frames) for a semantic action (e.g., run) across species, before and after spectral basis alignment.

Table 3: User study comparison on text alignment, 3d alignment and motion quality.

| Method | Text Align. | 3D Align. | Motion |
|---|---|---|---|
| Animate3d (Jiang et al., 2024) | 2.08 | 1.75 | 2.00 |
| AnimateAnyMesh (Wu et al., 2025) | 2.57 | 1.92 | 2.60 |
| SpectralFlow (**Ours**) | **4.62** | **4.52** | **4.53** |

noting that the animations more accurately captured the actions described in the text prompts. These findings underscore the effectiveness of our approach in conveying motion semantics and improving alignment between text and motion.

### A.4 ADDITIONAL COMPARISON VISUALIZATIONS

To further support our comparisons, we provide additional qualitative results in Fig. 11. These examples demonstrate the consistency, semantic alignment, and geometric fidelity of SpectralFlow across a range of animal meshes and motion prompts. Compared to other state-of-the-art methods, our approach consistently achieves superior structure-aware motion generation and closer alignment with the given actions.

We further compared the quality of motion generation in 3D space. Since our task involves 4D generation without ground-truth geometry for evaluation, we employ the ARAP metric (Sorkine & Alexa, 2007) to measure them by computing the ARAP difference between all subsequent frames and the first frame in each sequence. This metric quantifies non-rigid deformation differences of local surfaces before and after deformation, where smaller values indicate less surface distortion and isometric discrepancy. Comparative results are presented in Figure 9 and Table 4. Although our error is larger than that of AnimateAnyMesh, this is due to AnimateAnyMesh producing relatively static or rigid motion results. We further demonstrate the deformation magnitude in Table 4 and Figure 9, calculated as the RMSE of point coordinates after rigid alignment of all other frames to the first frame. Smaller values indicate lower deformation magnitude. Our results show significantly higher deformation magnitude than AnimateAnyMesh, validating the weak generalization capability of AnimateAnyMesh. Additionally, we tested these values on randomly sampled ground-truth sequences, revealing that our numerical distribution aligns more closely with ground-truth while maintaining minimal surface distortion.

### A.5 ADDITIONAL RESULTS OF SPECTRALFLOW

To further evaluate the generalization ability of our method, we provide additional qualitative results in Fig. 12. Specifically, we showcase reconstruction outputs on shapes generated by the Image-to-3D pipeline and from Animal3d dataset (Fig. 12). All examples are **unseen** shapes or categories (e.g., species) that were not present in the training data. The consistent reconstruction quality across

Table 4: Comparisons of deformation degree (RMSE) and ARAP loss of generated mesh sequences.

| Method | RMSE | ARAP |
|---|---|---|
| AnimateAnyMesh (Wu et al., 2025) | 1.52e-7 | 0.015 |
| SpectralFlow (**Ours**) | 2.13e-4 | 0.072 |
| Ground-Truth | 4.23e-4 | 0.131 |

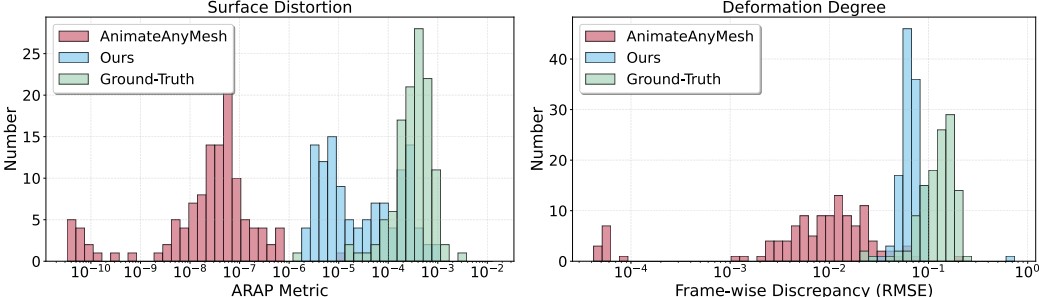

Figure 9: Visualizations of deformation degree (RMSE) and ARAP loss of generated mesh sequences.

diverse inputs demonstrates the strong generalization capability of SpectralFlow to novel and out-of-distribution shapes.

### A.6 VISUALIZATION OF THE EFFECT OF $\lambda_{\text{LAP}}$ ON RECONSTRUCTION

To illustrate the effect of the Laplacian regularization term $\lambda_{\text{lap}}$ in Eq. 3, we visualize the reconstructed meshes obtained by solving for the spectral coefficients $\hat{C}$ under different values of $\lambda_{\text{Lap}}$. As shown in Fig. 10, increasing $\lambda_{\text{Lap}}$ produces smoother reconstructions, especially when the input mesh $\mathcal{M}_0$ is noisy or of low quality. A clear example can be observed in the second row, second column of Fig. 10, where the legs of the horse become significantly smoother with a larger $\lambda_{\text{Lap}}$. This demonstrates the effectiveness of Laplacian regularization in suppressing high-frequency noise and improving the surface quality of the reconstructed shape. In our method, we set $\lambda_{\text{lap}} = 1e^{-3}$ to balance reconstruction fidelity and smoothness in most cases.

### A.7 DATA REPROCESSING AND NORMALIZATION

To ensure consistent and robust performance across sequences and species, we applied the following data preprocessing steps: After aligning the spectral coefficients $\hat{C}_t$, we performed global normalization across all sequences and species using the global mean and standard deviation. This step standardizes the scale of spectral features across different species, facilitating more stable training and evaluation.

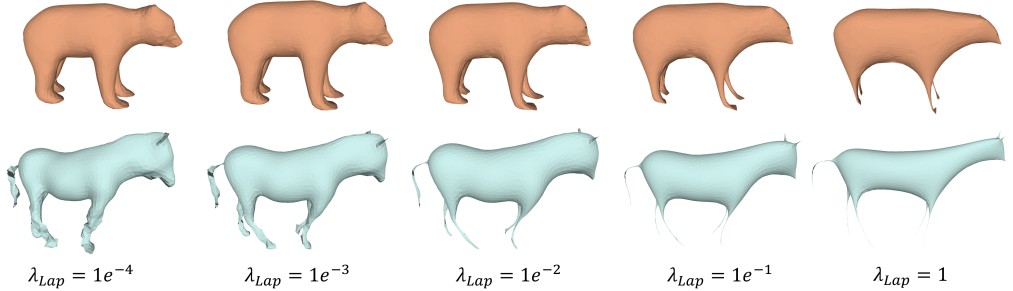

$\lambda_{Lap} = 1e^{-4}$     $\lambda_{Lap} = 1e^{-3}$     $\lambda_{Lap} = 1e^{-2}$     $\lambda_{Lap} = 1e^{-1}$     $\lambda_{Lap} = 1$

Figure 10: Visualization of reconstruction results under different $\lambda_{\text{Lap}}$ values. A larger value of $\lambda_{\text{Lap}}$ encourages smoother mesh surfaces. Appropriately setting $\lambda_{\text{Lap}}$ helps maintain surface regularity, particularly when the input mesh is noisy or of low quality.

Rendered results of 3D animations

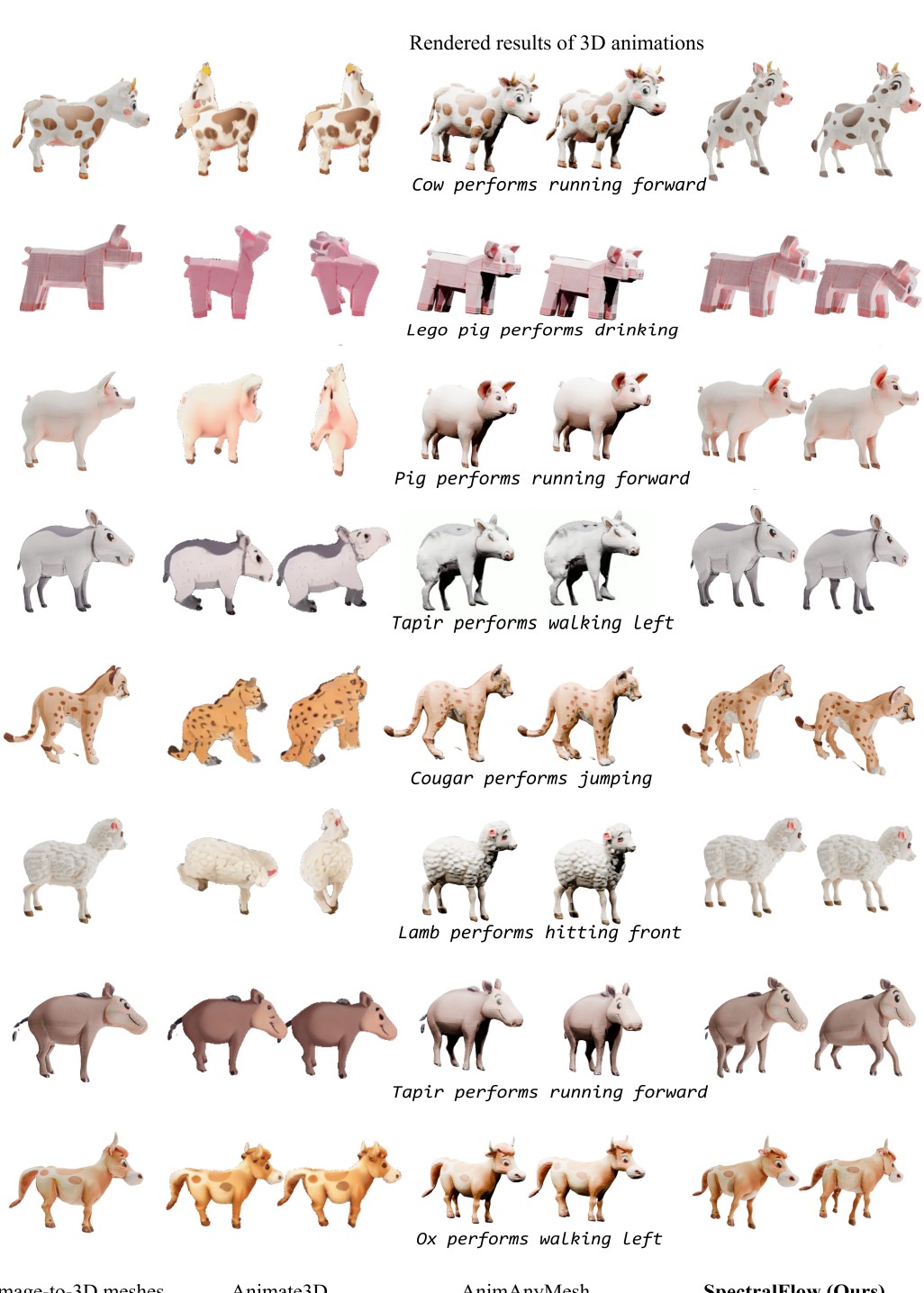

Image-to-3D meshes        Animate3D        AnimAnyMesh        **SpectralFlow (Ours)**

Figure 11: Additional comparison visualizations with state-of-the-art methods. Our method preserves motion semantics across diverse shapes by learning shared motion representations that generalize well across different species.

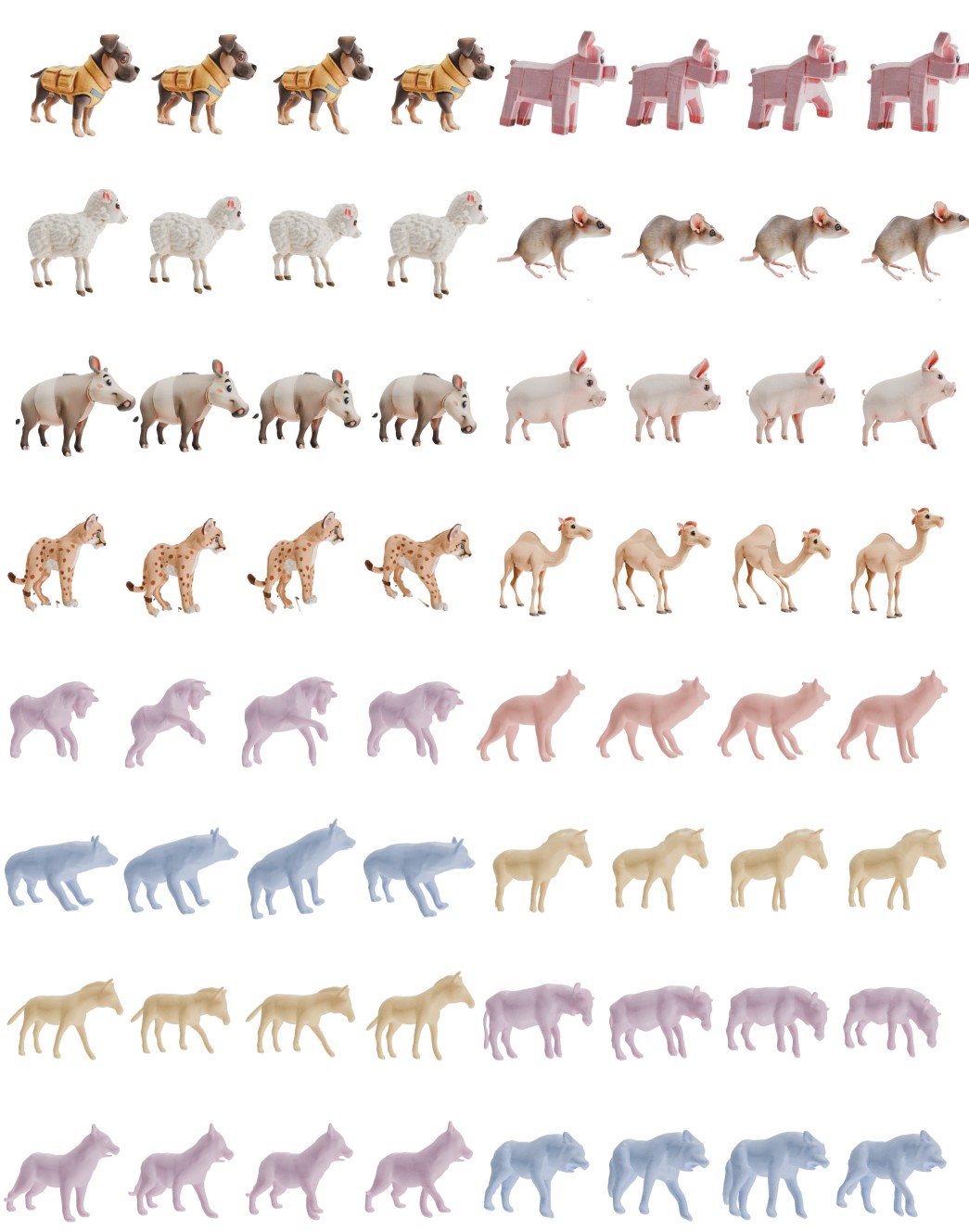

Figure 12: Additional animation examples of SpectraFlow. Our method generates realistic and coherent motion sequences for meshes obtained from Image-to-3D (Li et al., 2024a) and Animal3D (Xu et al., 2023) (via mesh extraction from images).

