# OpenReview forum: "SpectralFlow: Geometry-Aware Mesh Animation via Spectral Coefficient Diffusion"
_ICLR.cc/2026/Conference — Submitted to ICLR 2026_

### Official Review · Reviewer_k9PP · 2025-10-27

**Soundness:** 3
**Presentation:** 3
**Contribution:** 2
**Rating:** 4
**Confidence:** 2

**Summary:**

The paper proposes SpectralFlow, a diffusion framework for action-conditioned 4D (mesh-sequence) generation that operates in the Laplacian spectral domain instead of raw vertex space. Each mesh is represented by a truncated set of eigenfunctions of the discrete Laplace–Beltrami operator and time-varying spectral coefficients; cross-shape learning is enabled by functional-map–based basis alignment and a sign-correction scheme to resolve eigenfunction ambiguities. Motion is learned as trajectories of spectral-coefficient offsets with a conditional DDPM guided by action labels and shape features (DiffusionNet). For reconstruction, the method solves a regularized least-squares problem with Laplacian and l2 terms, yielding a closed-form solution; a deformation-graph propagation step enhances high-frequency details at inference. Experiments on Image-to-3D and Animal3D report improvements on VBench metrics and user study preferences over Animate3D/AnyMesh.

**Strengths:**

1. For motivation, learning in a shared spectral space with functional-map alignment and sign disambiguation elegantly addresses cross-shape consistency.
2. For method, the reconstruction energy and closed-form solution are explicit; temporal smoothness and conditioning are straightforward.
3. Metrics on VBench and user preference across two evaluation sets show some gains.

**Weaknesses:**

1. The alignment depends on functional maps and soft correspondences; failure modes (species with weak correspondences, topological artifacts) and sensitivity to noisy templates are not systematically evaluated.
2. There are no ablation studies in SpectralFlow, which should be added to illustrate the core component of the method.

**Questions:**

1. Though following the evaluation setting of Animate3D and AnimateAnyMesh, I'm still curious about if authors can test on mesh-space or correspondence-space metric (e.g., per-vertex/part consistency, surface distortion, isometry deviation).

I'm not an expert in this area, so please kindly answer the questions and weakness raised. I'll raise my score for proper answer.-

---

> ### Author Response · Authors · 2025-11-24
> **Response to Reviewer k9PP**
>
> Thank you for the reviewer’s valuable suggestions. Below, we provide our responses to the questions and concerns raised.
>
>     The alignment depends on functional maps and soft correspondences; failure modes (species with weak correspondences, topological artifacts) and sensitivity to noisy templates are not systematically evaluated.
> **Response:**
> Our method achieves robust motion generation even with inaccurate or partial correspondences, by aligning shapes in low-frequency spectral space. This approach tolerates non-isometric deformations and incomplete geometry, as shown in **Figure 6**. Some of these challenge species can be found on our latest [**Project page**](https://specflow3d.github.io./). Our method is robust to correspondence errors, as successful motion generation only requires approximate alignment in the low-frequency spectral space.
>
>     There are no ablation studies in SpectralFlow, which should be added to illustrate the core component of the method.
> **Response:**
> To evaluate the contribution of each component, we conducted comprehensive ablation experiments, and the results are summarized in **Table 2** and **Figure 7** of the updated manuscript.
> We included ablations on the number of spectral coefficients $k$, showing that $k$ = 32 offers a good balance between motion expressiveness and geometric fidelity.
> We also constructed a vertex-based baseline without spectral representation, which fails to generate coherent or stable motion, underscoring the advantage of our compact and geometry-aware representation.
> Moreover, when training on only 50% or 25% of the data, our model still captures meaningful motion patterns, thanks to the strong geometric guidance provided by the spectral representation.
>
>     Though following the evaluation setting of Animate3D and AnimateAnyMesh, I'm still curious about if authors can test on mesh-space or correspondence-space metric (e.g., per-vertex/part consistency, surface distortion, isometry deviation).
> **Response:**
> We thank the reviewer for the suggestion. While 2D video-based evaluation remains the primary metric for current 4D generation benchmarks, as it reflects perceptual motion quality and is widely used in works like Animate3D and AnimateAnyMesh.
> Since our task involves 4D generation without ground-truth geometry for evaluation, we employ the ARAP loss [Sorkine et al. 2007] to measure motion quality in 3D space by computing the ARAP difference between all subsequent frames and the first frame in each sequence. This metric quantifies non-rigid deformation differences of local surfaces before and after deformation, where smaller values indicate less surface distortion and isometric discrepancy. Comparative results are presented in **Figure 9** and **Table 4**. Although our error is larger than that of AnimateAnyMesh, this stems from AnimateAnyMesh producing relatively static or rigid motion results. We further demonstrate the deformation magnitude in **Table 4** and **Figure 9**, calculated as the RMSE of point coordinates after rigid alignment of all other frames to the first frame. Smaller values indicate lower deformation magnitude. Our results show significantly higher deformation magnitude than AnimateAnyMesh, validating the weak generalization capability of AnimateAnyMesh. Additionally, we tested the metric distribution on ground-truth sequences, revealing that our numerical distribution aligns more closely with ground-truth while maintaining minimal surface distortion.
> Note that Animate3D is not included in the 3D-space geometric evaluation, as it is designed for multi-view video generation rather than direct 4D content prediction.
>
> [Sorkine et al. 2007] Sorkine et al., As-rigid-as-possible surface modeling, Symposium on Geometry processing, 2007
>
> Table 2. Ablation study. Lower values indicate better performance across all metrics.
> | **Metrics**                | **k=16** | **k=64** | **Full Vetex** | **25%Data** | **50%Data** | **Ours**     |
> |---------------------------|----------|----------|----------------|-------------|-------------|--------------|
> | Diffusion Loss$\downarrow$     | 0.0085   | 0.0092   | 0.7860         | 0.0263      | 0.0150      | **0.0062**   |
> | Vetex Error$\downarrow$        | 0.1466   | 0.1850   | 0.3474         | 0.1262      | 0.1049      | **0.0583**   |
> | Smoothness Loss$\downarrow$   | 0.0178   | 0.0169   | 28.560         | 0.0221      | 0.0180      | **0.0101**   |
>
> Table 4. Comparisons of deformation degree (RMSE) and ARAP loss of generated mesh sequences.
> | Method           | **RMSE**  | **ARAP** |
> |------------------|-----------|----------|
> | AnimateAnyMesh   | 1.52e-7   | 0.015    |
> | Ours             | 2.13e-4   | 0.072    |
> | Ground-Truth     | 4.23e-4   | 0.131    |

---

> ### Author Response · Authors · 2025-11-26
> **Looking forward to further feedback**
>
> Dear Reviewer **k9PP**
>
> Thank you for taking the time to review our manuscript and for your valuable feedback and recognition. We have carefully addressed all the comments and concerns raised, as reflected in our detailed responses and the revised manuscript, supplementary material, and the anonymous project page.
>
> We are looking forward to hearing from you again.
>
> Best regards,
>
> The Authors

---

### Official Review · Reviewer_zU3q · 2025-10-29

**Soundness:** 3
**Presentation:** 3
**Contribution:** 2
**Rating:** 6
**Confidence:** 5

**Summary:**

The method proposes a novel method for geometry-aware mesh animation which generates realistic motion sequences for a variety of 3D shapes conditioned on action labels. It can also generate realistic motion sequences for unseen shapes that demonstrates strong generalization in both categories and actions. The key contribution of the method is to use a set of Laplacian eigenbases and a sequence of time-varying spectral coefficients to represent the motion sequence. A conditional diffusion model is trained to generate the time-varying spectral coefficients for geometry-aware mesh animation. Extensive experiments show that the proposed method outperforms prior methods in reconstruction quality and motion generalization.

**Strengths:**

1. The paper is well-written and easy to follow. The motivation of using time-varying spectral coefficients to represent motion sequence is well illustrated.
2. The overall framework is clear and efficient. The method trains a conditional diffusion model to predict time-varying spectral coefficients, which is more efficient and generalizable compared to vertex-wise deformation prediction. In order to better model the motion sequence across different shapes, the method proposes spectral basis alignment and sign correction across different shapes based on an existing shape matching method.
3. The paper also proposes an efficient and robust closed-form solution for the computation of an optimal spectral coefficients given ground-truth target shape, which is important for training a diffusion model.
4. Compared to baseline methods, the proposed method achieves better text alignment, 3d alignment and motion quality.

**Weaknesses:**

1. The proposed method solely uses spectral coefficients and the eigenbases of the input shape to represent motion sequence, which limits its ability to represent large motions such as large articulations.
2. As shown in the videos provided in the project website, the proposed method cannot fully decompose the pose-dependent motion and shape-dependent motion. Therefore, it also changes the shape of the object during the animation.

**Questions:**

1. How robust is the method against the shape matching method for spectral basis transformation? If the soft point-wise correspondences from Cao et al. 2023 are wrong, would it impact the final performance?

---

> ### Author Response · Authors · 2025-11-24
> **Response to Reviewer zU3q**
>
> Thank you for the reviewer’s valuable suggestions. Below, we provide our responses to the questions and concerns raised.
>
>     The proposed method solely uses spectral coefficients and the eigenbases of the input shape to represent motion sequence, which limits its ability to represent large motions such as large articulations.
> **Response:**
> We would like to clarify that our spectral representation **does not** limit the ability to represent large motions or articulations. The number of spectral coefficients ($k=32$) only affects the spatial frequency resolution, not the magnitude of motion. In practice, our method captures full-body movements such as walking, running, and turning without issue. We refer the reviewer to visit our updated [**Project page**](https://specflow3d.github.io.) for visual results demonstrating natural motions.
> What may be affected are fine-grained geometric details, such as subtle deformations in ears, tails, or fur-like structures, which are represented by high-frequency spectral components. These details are often secondary to the overall motion and can be preserved through deformation graph-guided reconstruction, as described in the
> last paragraph of **Section 3**.
>
>     As shown in the videos provided in the project website, the proposed method cannot fully decompose the pose-dependent motion and shape-dependent motion. Therefore, it also changes the shape of the object during the animation.
> **Response:**
> This problem arose because our training data was processed using **frame-wise** normalization, which introduced inconsistent scale variations across frames within a sequence. We have now resolved this problem and updated all test results accordingly. As shown in our [**Project page**](https://specflow3d.github.io./), the scale variation artifacts have been eliminated.  Our method achieves pose-shape separation by representing each sequence using a shared spectral basis (capturing shape), spectral coefficients sequences (capturing pose). Since the deformations are nearly isometric, the eigenbasis remains consistent across frames, enabling temporally coherent motion modeling.
>
>     How robust is the method against the shape matching method for spectral basis transformation? If the soft point-wise correspondences from Cao et al. 2023 are wrong, would it impact the final performance?
> **Response:**
> It is worth emphasizing that we focus on **matching the basis functions** rather than all vertices on the surface. Matching the basis is more tractable than point-wise matching, since the variables consist of only a 32×32 matching matrix.
> Our method achieves robust motion generation even with inaccurate or partial correspondences, by aligning shapes in low-frequency spectral space. This approach tolerates non-isometric deformations and incomplete geometry, as shown in **Figure 6**. Some of these challenge species can be found on our latest [**Project page**](https://specflow3d.github.io./). Our method is robust to correspondence errors, as successful motion generation only requires approximate alignment in the low-frequency spectral space.

---

> ### Author Response · Authors · 2025-11-26
> **Looking forward to your further feedback**
>
> Dear Reviewer **zU3q**
>
> Thank you for taking the time to review our manuscript and for your valuable feedback and recognition. We have carefully addressed all the comments and concerns raised, as reflected in our detailed responses and the revised manuscript, supplementary material, and the anonymous project page.
>
> We are looking forward to hearing from you again.
>
> Best regards,
>
> The Authors

---

### Official Review · Reviewer_9JJ1 · 2025-10-31

**Soundness:** 3
**Presentation:** 3
**Contribution:** 2
**Rating:** 4
**Confidence:** 4

**Summary:**

This paper proposes a method to animate 3d mesh. Instead of directly generating new mesh sequence, it proposes to generate the sequence in spectral space with diffusion models. To ensure this method works well, the paper also proposes a alignment method to establishe a shared spectral space for various topology.

**Strengths:**

1. Generate sequence in spectral domain can help to reduce the problems of previous methods such as high dimensionality and reliance on VAE latents. The method is novel.

2. The proposed method provides visualization and analysis of the results and distributions.

3. Instead of naively using the spectral domain, the proposed method also introduces spectral basis alignment and sign correction scheme to improve.

**Weaknesses:**

1. My biggest concern is the ablation part. This paper does not include much ablation experiments. For example, although some information can be found from the method part. The paper does not mention explicitly what is the dimension reduce factor. (e.g. the original length vs. k=32).

2. What if we train the model with naive mesh representation? Is it too high dimensional and can not be used? How is this naive method compared with the proposed method in terms of generation quality and speed.

3. Some of the motion visualization still does not seem very good based on the link provided. For example the cameral in the last video.

**Questions:**

The questions are included in the weaknesses part.

---

> ### Author Response · Authors · 2025-11-24
> **Response to Reviewer 9JJ1**
>
> Thank you for the reviewer’s valuable suggestions. Below, we provide our responses to the questions and concerns raised.
>
>     My biggest concern is the ablation part. This paper does not include much ablation experiments. For example, although some information can be found from the method part. The paper does not mention explicitly what is the dimension reduce factor. (e.g. the original length vs. k=32).
>
> **Response:**
> We have included ablation studies with different configurations (e.g., $k = 16, k = 64$, and full vertex representation), as shown in **Table 2** and **Figure 7** of the updated PDF file.
> **Table 2** shows that $k = 16$ lacks geometric detail, while $k = 64$ increases the challenge of matching accuracy and overfits to high-frequency variations and negatively impacts motion learning. Our default setting of $k=32$ provides a favorable balance, capturing essential motion while preserving geometric fidelity.
>
>     What if we train the model with naive mesh representation? Is it too high dimensional and can not be used? How is this naive method compared with the proposed method in terms of generation quality and speed.
> **Response:**
> We evaluate a vertex-based baseline that removes the spectral representation and learns directly from raw vertex positions. As shown in **Figure 7** (the fourth column), it fails to produce coherent motion or stable shape sequences, as excessively high dimensions increase the difficulty of motion learning. Compact motion representation is crucial for capturing shared motion semantics across diverse species.
>
>     Some of the motion visualization still does not seem very good based on the link provided. For example the cameral in the last video.
> **Response:**
> This problem arose because our training data was processed using frame-wise normalization, which introduced inconsistent scale variations across frames within a sequence. We have now resolved this problem and updated all test results accordingly. We kindly refer the reviewer to visit our [**Project page**](https://specflow3d.github.io./), the scale variation artifacts have been solved.
>
> Table 2: Ablation study. Lower values indicate better performance across all metrics.
> | **Metrics**                | **k=16** | **k=64** | **Full Vetex** | **25%Data** | **50%Data** | **Ours**     |
> |---------------------------|----------|----------|----------------|-------------|-------------|--------------|
> | Diffusion Loss$\downarrow$     | 0.0085   | 0.0092   | 0.7860         | 0.0263      | 0.0150      | **0.0062**   |
> | Vetex Error$\downarrow$        | 0.1466   | 0.1850   | 0.3474         | 0.1262      | 0.1049      | **0.0583**   |
> | Smoothness Loss$\downarrow$   | 0.0178   | 0.0169   | 28.560         | 0.0221      | 0.0180      | **0.0101**   |

---

> ### Author Response · Authors · 2025-11-26
> **Looking forward to your further feedback**
>
> Dear Reviewer **9JJ1**
>
> Thank you for taking the time to review our manuscript and for your valuable feedback. We have carefully addressed all the comments and concerns raised, as reflected in our detailed responses and the revised manuscript, supplementary material, and the anonymous project page.
>
> We are looking forward to your further feedback.
>
> Best regards,
>
> The Authors

---

### Official Review · Reviewer_8WcF · 2025-11-01

**Soundness:** 2
**Presentation:** 2
**Contribution:** 2
**Rating:** 4
**Confidence:** 3

**Summary:**

This paper introduces SpectralFlow, a diffusion-based framework for generating 4D mesh animations conditioned on an input mesh and an action label. The core idea is to operate in the Laplacian spectral domain rather than on raw vertex coordinates. The method first computes a lowfrequency laplacian eigenbasis for the input shape. To enable learning across different shapes and topologies, it aligns this basis to a canonical template using a transformation matrix and a sign correction matrix , both derived from a pretrained functional map based correspondence network. Motion is then represented as a trajectory of spectral coefficients, modeled as an offset from the input's canonical pose. A Transformer based diffusion model is trained to generate these spectral coefficient trajectories, conditioned on the shape's spectral features and the action label. Finally, a deformation graph based method propagates the resulting low frequency motion back to the original high-resolution mesh. The method is evaluated on animal datasets, reporting strong generalization to unseen shapes.

**Strengths:**

1 using a diffusion model to generate trajectories of laplacian spectral coefficients is a novel approach. It leverages a compact, intrinsic, and low dimensional representation of shape, which is different from common vertex-based or implicit field-based methods.
2. operating in a very low dimensional space (k=32 coefficients per frame) instead of on thousands of vertices, the method significantly reduces the dimensionality of the generation problem.

**Weaknesses:**

1. the method only models the first k=32 low-frequency eigenfunctions. This is a strong low-pass filter on motion, making it fundamentally incapable of generating high-frequency details. The final "smooth" look is a limitation of the representation.
2. the framework is not an end-to-end generator. The pipeline is quite complex and might introduce noises when generalizing to more complex motions.
3. the visual results have clear artifacts that the entire shape varies in scale durong the motion.
4. current results only show limited motion and yet not smooth
5. even the authors claim they can generalize to various unseen shapes but the unseen shapes are all quadrepeds with similar topology.

**Questions:**

1 The authors suggest this low-dimensional space is more data efficient. Have you performed any experiments to validate this claim directly? For example, how does the model's performance degrade when trained on a smaller fraction (50% or 25%) of the training data?

---

> ### Author Response · Authors · 2025-11-24
> **Response to Reviewer 8WcF**
>
> Thank you for the reviewer’s valuable suggestions. Below, we provide our responses to the questions and concerns raised.
>
>     The method only models the first k=32 low-frequency eigenfunctions. This is a strong low-pass filter on motion, making it fundamentally incapable of generating high-frequency details. The final "smooth" look is a limitation of the representation
>
> **Response:**
> *Firstly*, unlike the geometric shape space, the motion space does not contain as many high-frequency details. Most motions, such as limb movements, are relatively low-frequency. Therefore, our method can effectively handle the majority of motion signals. By mapping motion information into a more compact space, we can more easily learn common motion patterns from limited datasets.
> *Secondly*, the 4D generation task is extremely challenging due to the severe scarcity of 4D datasets.
> Although works like AnimateAnyMesh demonstrate impressive demos, in testing, most unseen shapes result in no motion or only minor motions (see **Figure 5** and **Figure 12** of the updated PDF file). For some examples that generated pronounced motion, there is a significant misalignment between the motion and the text condition (see **Figure 5** and **Figure 12**). By contrast, our motion generation is more robust and exhibits better semantic alignment.
> *Finally*, we aim to integrate more geometric priors to enhance generalization in 4D generation with limited data. While our method cannot generate high-frequency details, such as toe movements, our coarse motions are already significantly better than those of other comparative methods and can serve as a foundational result for further fine-grained motion generation.
>
>
>     The framework is not an end-to-end generator. The pipeline is quite complex and might introduce noises when generalizing to more complex motions.
> **Response:**
> Common techniques for 4D generation, such as AnimateAnyMesh[Wu et al. 2025] and DNF[Zhang et al. 2025], typically require additional training of a VAE to construct a latent representation for diffusion model training. Our method effectively replaces the VAE training process with spectral decomposition and matching operations. These operations are computationally efficient, taking approximately 4 seconds. Thus, although not end-to-end, our approach **does not incur significant additional overhead**. Our matching module incorporates a well-established technique from shape matching. As this open research field continues to advance, it is expected to achieve better performance and increased robustness against more complex noise data.
>
>
>     The visual results have clear artifacts that the entire shape varies in scale during the motion.
> **Response:**
> This problem arose because our training data was processed using **frame-wise** normalization, which introduced inconsistent scale variations across frames within a sequence. We have now resolved this problem and updated all test results accordingly. We kindly refer the reviewer to visit our [**Project page**](https://specflow3d.github.io./), where the scale variation artifacts have been eliminated.
>
>     Current results only show limited motion and yet not smooth even the authors claim they can generalize to various unseen shapes but the unseen shapes are all quadrepeds with similar topology.
> **Response:**
> As mentioned above, we have resolved the scale variation issue and now demonstrate smoother results. Given that our training dataset comprises only quadrupedal animals, it is nearly impossible to generalize to other topological structures. Nevertheless, even within quadrupeds, significant challenges exist, such as handling the differences between a mouse and a camel. We are able to process such partially matched cases, as illustrated in **Figure 6**.
> Furthermore, as the dataset contains more types of objects, we will be able to handle more species.
>
>     The authors suggest this low-dimensional space is more data efficient. Have you performed any experiments to validate this claim directly? For example, how does the model's performance degrade when trained on a smaller fraction (50% or 25%) of the training data?
> **Response:**
> For the methods being compared, AnimateAnyMesh utilizes 66,209 sequences, Animate3D employs 115,566 sequences, whereas our method requires only **1,700** sequences. Despite this, our approach demonstrates significantly superior generalization performance compared to these methods, as evidenced in **Figure 5**, **Figure 12**, **Table 1**, and the [**Project page**](https://specflow3d.github.io./). Thus, our method is more data-efficient. Additionally, we evaluated performance under even more limited training data (**Section 4**. (Ablation Study) of the updated PDF file), where our method remains trainable and yields reasonably competitive results.

---

> ### Author Response · Authors · 2025-11-26
> **Looking forward to your further assessment**
>
> Dear Reviewer **8WcF**
>
> Thank you for taking the time to review our manuscript and for your valuable feedback. We have carefully addressed all the comments and concerns raised, as reflected in our detailed responses and the revised manuscript, supplementary material, and  the anonymous project page.
>
> We are looking forward to your further assessment.
>
> Best regards,
>
> The Authors

---

### Author Response · Authors · 2025-11-24
**General Response**

We thank all reviewers for their time and constructive comments.  We have carefully reviewed the comments and made revisions accordingly. We have updated the revised version and highlighted the modifications in red.
Here, we summarize the key concerns:

- **Contribution**
We propose a novel spectral representation for 4D modeling that encourages the model to understand motion semantics rather than overfit to motion trajectories.
Due to the severe scarcity of 4D data, training purely implicit generative models directly presents significant challenges. Therefore, we aim to incorporate more 3D shape information, design low-dimensional representations, and obtain **compact motion vectors** to achieve better generation performance with limited data.
Our method captures shared semantic motions across species, enabling robust generalization beyond individual shapes or motion sequences.

- **Performance**
SpectralFlow not only generates **structure-aware motion** across species with high consistency and fidelity, but also preserves **motion semantics** remarkably well, leading to clearly superior overall performance.
Although our method performs well with limited 4D data, it focuses on capturing coarse motion patterns and may overlook finer-grained details.
We highlight generalization to novel shapes and motions as a key strength, and consider richer dynamics and longer sequences as promising directions for future work.

- **Dependence of correspondences**
Our method relies on correspondences extracted by existing techniques; however, our matching approach differs from the point-wise dense matching processed by shape matching. We only need to **establish matches between a small number of basis functions**, requiring only a matching matrix of size $k\times k$ ($k=32$ by default). Consequently, the complexity is significantly reduced. Additionally, partially inaccurate points do not adversely affect our results, as shown in **Figure 6** of the updated PDF file.

- **Ablation study**
We added ablation studies including the selection of $k$, naive mesh representation, and the size of training data in Sec. 4 of the updated PDF file.

We appreciate the reviewers' insights, which significantly helped us refine and strengthen our work.

---

### Author Response · Authors · 2025-11-25
**Looking forward to your further assessment**

Dear **Reviewers**,

Thank you for taking the time to review our manuscript and for your valuable feedback and recognition. We have carefully addressed all the comments and concerns raised, as reflected in our detailed responses and the revised manuscript and supplementary material.

We sincerely appreciate your efforts and look forward to your further assessment.

Best regards,

The Authors

---

> ### Author Response · Authors · 2025-11-27
> **Looking forward to your further feedback**
>
> Dear **Reviewers of Paper 4411**,
>
> Have our comprehensive responses addressed your concerns? We are looking forward to your further feedback.
>
> Best regards,
>
> The Authors

---

### Author Response · Authors · 2025-11-29
**Summary for the Area Chair**

Dear **Area Chair**,

We sincerely hope the following summary will help the new area chair quickly grasp the main contributions and our responses.

We sincerely thank the reviewers for their time and constructive feedback. Notably, none of the reviewers raised concerns regarding the **novelty** or conceptual contribution of our work [*8WcF*, *9JJ1*, *zU3q*, *k9PP*]. Reviewers acknowledged **the novelty of our spectral representation** for 4D generation [*8WcF*, *9JJ1*]. They highlighted the **compactness and efficiency of our low-dimensional representation**, and its effectiveness compared to vertex-based deformation or vae-latent generation [*8WcF*, *9JJ1*, *zU3q*]. The proposed spectral basis alignment, sign correction, and closed-form coefficient recovery were also recognized as **effective and well-motivated** [*9JJ1*, *zU3q*, *k9PP*]. The paper was considered clear and well-structured, with **strong performance** across standard benchmarks [*zU3q*, *k9PP*].

**The concerns were primarily related to experimental aspects**, including ablation studies [*8WcF*, *9JJ1*, *k9PP*], robustness to correspondence [*zU3q*, *k9PP*], scale artifacts [*8WcF*, *9JJ1*, *zU3q*], and evaluation metrics [*k9PP*].
**We have addressed all these concerns thoroughly in our rebuttal**, including new experiments and addressing all reported issues. The **updated manuscript** (with revisions marked in red) and the **[latest Project Page](https://specflow3d.github.io./)** reflect the improvements and demonstrate the effectiveness of our method.

We hope this summary helps the area chair quickly understand the contributions and our responses. We believe the paper is now in a strong position for acceptance and will make a meaningful contribution to the community. We greatly appreciate the Area Chair’s time and effort in handling this submission.


Best regards,

The Authors

---

### Meta-Review · Area_Chair_ehjw · 2026-01-07

**Summary:**

This work received three marginally below ratings and one marginally above rating. The author rebuttals have addressed to a certain degree the issues of missing ablation study, correspondence, scale artifacts. However, it shows only smooth motions, and the pipeline is quite complex, not an end-to-end framework, both of which make the method hard to generalize to complex motions. All the shapes are also limited to quadrupeds with similar topology. ACs find this work as a real borderline case, having pros and cons, and do not recommend acceptance in its present form considering the competitiveness of this venue.

**Reviewer Concerns:**

The same as above.

**Reviewer Scores:**

This work received three marginally below ratings and one marginally above rating. The author rebuttals have addressed to a certain degree the issues of e.g. missing ablation study.

---

### Decision · Program_Chairs · 2026-01-26

Reject